# Wearable Technologies for Mental Workload, Stress, and Emotional State Assessment during Working-Like Tasks: A Comparison with Laboratory Technologies

**DOI:** 10.3390/s21072332

**Published:** 2021-03-26

**Authors:** Andrea Giorgi, Vincenzo Ronca, Alessia Vozzi, Nicolina Sciaraffa, Antonello di Florio, Luca Tamborra, Ilaria Simonetti, Pietro Aricò, Gianluca Di Flumeri, Dario Rossi, Gianluca Borghini

**Affiliations:** 1BrainSigns, SRL, 00185 Rome, Italy; vincenzo.ronca@uniroma1.it (V.R.); alessia.vozzi@uniroma1.it (A.V.); nicolina.sciaraffa@uniroma1.it (N.S.); antonello.diflorio@brainsigns.com (A.d.F.); pietro.arico@uniroma1.it (P.A.); gianluca.diflumeri@uniroma1.it (G.D.F.); dario.rossi@brainsigns.com (D.R.); gianluca.borghini@uniroma1.it (G.B.); 2Department of Anatomical, Histological, Forensic and Orthopaedic Sciences, Sapienza University, 00185 Rome, Italy; luca.tamborra@it.ey.com (L.T.); ilaria.simonetti@it.ey.com (I.S.); 3Department of Molecular Medicine, Sapienza University of Rome, 00185 Rome, Italy; 4Ernst & Young, Department People Advisory Services, 00187 Rome, Italy; 5IRCCS Fondazione Santa Lucia, 00179 Rome, Italy; 6Department of Business and Management, LUISS University, 00197 Rome, Italy

**Keywords:** wearable device, emotional state, mental workload, stress, heart rate, eye blinks rate, skin conductance level

## Abstract

The capability of monitoring user’s performance represents a crucial aspect to improve safety and efficiency of several human-related activities. Human errors are indeed among the major causes of work-related accidents. Assessing human factors (HFs) could prevent these accidents through specific neurophysiological signals’ evaluation but laboratory sensors require highly-specialized operators and imply a certain grade of invasiveness which could negatively interfere with the worker’s activity. On the contrary, consumer wearables are characterized by their ease of use and their comfortability, other than being cheaper compared to laboratory technologies. Therefore, wearable sensors could represent an ideal substitute for laboratory technologies for a real-time assessment of human performances in ecological settings. The present study aimed at assessing the reliability and capability of consumer wearable devices (i.e., Empatica E4 and Muse 2) in discriminating specific mental states compared to laboratory equipment. The electrooculographic (EOG), electrodermal activity (EDA) and photoplethysmographic (PPG) signals were acquired from a group of 17 volunteers who took part to the experimental protocol in which different working scenarios were simulated to induce different levels of mental workload, stress, and emotional state. The results demonstrated that the parameters computed by the consumer wearable and laboratory sensors were positively and significantly correlated and exhibited the same evidences in terms of mental states discrimination.

## 1. Introduction

This paper aims to investigate the capability of two consumer wearable devices (i.e., Empatica 4 and Muse 2) in assessing different levels of mental and emotional states. The consumer devices were compared to laboratory ones (i.e., BeMicro and Shimmer) in order to validate their reliability in scientific research.

### 1.1. Monitoring Mental States

In recent years there was an increasing interest toward wearable monitoring devices to assess physiological and mental activity, both in research and industry [1,2]. These devices are particularly important to the world’s increasingly aging population since this aspect constitutes a relevant risk factor for work-related accidents [3]. Both in research and industry domains the mental states’ monitoring is becoming really important. Starting from few decades ago, there was a shift in the focus from operators’ physical demands to their cognitive demands. This shift is particularly evident for some complex and safety-critical human activities such as air traffic control, and car and rail train driving [4,5,6,7]. In these contexts, it is evident that most of the fatal and non-fatal accidents occur because of Human Factors (HFs) concerns [8,9,10,11]. Among all the HFs, stress, mental overload, and lack of vigilance could cause tragic human errors in several working environments [12,13,14]. Giving the limitations imposed by subjective evaluation of mental states [15,16,17] and due to the fact that in some specific activities it is not possible to interrupt operators while working, researchers started to acquire biosignals to monitor and assess operators’ mental states. Biomarkers such as skin conductance level (SCL), heart rate (HR), and eye blink rate (EBR) are investigated as correlates of users’ mental states to develop a monitoring system to diminish and prevent fatal and non-fatal accidents [4,6,16,18,19,20]. For this reason, it is important to reduce at minimum the invasiveness of the monitoring equipment. Furthermore, the interest in consumer wearable devices was supported by the increasing advances in microelectronics which allowed to overcome the limitations imposed by the size of the electronic components and of the measuring sensor itself [21]. The size reduction, other than costs reduction and easiness to use, enhanced the application of such wearable devices to areas of research which were usually investigated using laboratory technologies, considered in scientific literature as the gold-standard [22,23,24]. Indeed, despite the improvements of the technology behind laboratory equipment such devices are often uncomfortable and obtrusive for the participants leading to a non-optimal condition to ecologically assess mental states [25].

### 1.2. Consumer Wearables in Scientific Research

Consumer wearable devices are ideal candidates to record operators’ biosignals without negatively interfere with their activities and tasks. Given the emergence of an incredible amount of commercial and user-friendly wearable devices [23,24] and given the fact that they seem to better adapt to daily-life activities, their accuracy has to be investigated deeply. The reliability of wearable devices in measuring biomarkers such as HR and SCL was demonstrated. In fact, compared to gold-standard equipment, consumer wearable devices showed a similar accuracy in measuring different biomarkers such as HR, HRV and SCL in different conditions [26,27,28]. Ragot and colleagues successfully adopted the Empatica E4 wrist-band to measure physiological response in an emotion recognition task [29]. Based on these evidence wearable devices were also used to assess different mental states. Setz and colleagues [30] compared the reliability of a consumer wearable device (Empatica E4) in detecting drowsiness during a driving simulation task using HRV. The authors found that E4 wristband showed similar results compared to a medical-grade device and argued that the latter device could be substituted with the E4 in order to detect drowsiness. The possibility to discriminate between different levels of the same mental states was also explored. A study on simulated train traffic controlling [25] demonstrated that it is possible to differentiate between different level of mental workload (WL) using HRV acquired via wearable device. Compared to an FDA-approved medical device authors showed that a consumer wearable sensor (EmWave Pro, Boulder Creek, California, USA) had similar results in estimating changes in HRV, while the Empatica E3, a different consumer wearable device included in the same study, did not show the same reliability. The potentiality of consumer wearable devices in acquiring biosignals in an unobtrusive way brought to the development of devices to collect electroencephalographic (EEG) and electrooculographic (EOG) signals. Krigolson and colleagues [31] validated Muse 2 wearable device for ERP research demonstrating an adequate level of accuracy in measuring N200 and P300 components compared to standard 10–20 electrode configuration. Other researchers investigated the possibility to use Muse 2 to discriminate between different levels of enjoyment while playing videogames [32]. The authors reported no significant difference in cortical activity while subjective reports did but the absence of a gold standard reference did not allow to objectively assess the accuracy of the consumer wearable EEG device considered.

### 1.3. Aim of the Present Study

Summarizing, there are contrasting evidence in literature about the reliability of consumer wearable devices. The possibility of these devices to estimate different biosignals is well accepted [26,27,28,29,32]. Additionally, some authors successfully differentiated between several mental states using the neurometrics collected with consumer wearables devices [25,30] but in other cases a failure was reported [25,31]. This paper fits into this contest by comparing the Empatica E4 and Muse 2 with laboratory equipment. The reliability and capability of the two consumer wearable devices were investigated for stress, mental workload (WL), and emotional state (EmS) evaluation while participants were performing three working-like tasks, comparing them with laboratory equipment. To summarize, this paper aimed at responding to the following research questions (RQ):RQ1: Are the above-mentioned neurophysiological parameters (EBR, SCL and HR) gathered through consumer wearable devices comparable with those acquired with laboratory equipment?RQ2: Are consumer wearable devices reliable in discriminating different levels of the mental states considered (WL, Stress and EmS)?

## 2. Materials and Methods

### 2.1. Participants

Seventeen (17) participants were recruited from the Sapienza University of Rome (ten males and seven females, 31.1 ± 3.7 years old) with normal or corrected-to-normal vision. Due to artifacts and missing data caused by technical issues after signals processing twelve (12) participants were considered valid for the analysis. Informed consent was obtained from each participant after explanation of the study. The experiment was conducted following the principles outlined in the Declaration of Helsinki of 1975, as revised in 2000 and was approved by the Sapienza University of Rome Ethical Committee in Charge for the Department of Molecular Medicine (protocol number: 2507/2020, approved on 4 August 2020). To respect the privacy of participants, only aggregate results were reported.

### 2.2. Procedures

In order to test the reliability of consumer wearable devices in WL, stress, and EmS evaluation, an experimental protocol was designed including three tasks: N-back task, Doctor Game task, and Webcall task. These tasks were selected to respectively simulate an office-like environment, an assembly-line and a teleworking activity. N-back task was used to simulate an office related activity which usually do not demand a pronounced physical effort whilst keeping high the mental one. Doctor Game (i.e., “Operation”) represents a fine motor skill task requiring participant to use a pair of tweezers to extract several items from their slots. This task was adopted because of its analogy with the assembly line activities. Finally, Webcall task was used to reproduce a teleworking case, in which people are often asked to communicate and coordinate with someone who is not physically present. The order of tasks completion was balanced and randomized among participants.

#### 2.2.1. N-Back Task

The N-back task (NB) (Figure 1) is a robust psychological test to manipulate working memory load [33], one of the major components and a reasonable approximation of WL [34]. Participants are presented with a sequence of letters on a screen. The goal is to press a button when the letters appearing on the screen is the same that occurred in the series *n* steps before. The difficulty of the task can be manipulated increasing the value of *n*, thus forcing participants to retain more items in their mind. In this study, the task was composed of a baseline and three conditions: Low WL, high WL, and stress. Under all conditions, 21 uppercase letters were used, which were displayed for 500 ms and an inter-stimulus interval randomized between 500 to 3000 ms; 33% of the displayed letters were targets.

Baseline: Participants were instructed to watch the sequence of letters without giving any response.Low WL: 0-back. The task consisted in indicating when the stimulus on the screen matches a predetermined letter.High WL: 2-back. The task consisted in indicating when the stimulus occurred in the series 2 steps before. When investigating stress assessment, we referred to this condition as ‘No Stress’ condition (i.e., in the comparison ‘No Stress vs. Stress’) as it differed from the Stress one only in the presence of the stressors whilst the difficulty level was the same.Stress: The task was practically equivalent to the High WL one (indicating when the stimulus occurred in the series two steps before) but simultaneously high intensity noise was played (85 db) and the white-coat effect was used to stress the participant [35]. Four-minute relaxing music and video was played at the end of this phase for letting the participants recover from the stressful event before continuing with the remaining experimental conditions [36].

In all conditions, behavioral data like reaction time and number of errors were collected. The low WL and the high WL conditions were performed randomly while the baseline and the stress conditions were performed respectively at the beginning and at the end of the experimental task. Before the 0-back and the 2-back task, the participant performed a training session containing 21 stimuli, 33% of which were targets.

#### 2.2.2. Doctor Game Task

This task is a fine motor skill task. We adopted the “Doctor Game” (DG) (i.e., “Operation”) board game (Figure 2). Its goal consisted in removing small objects from the board without touching the metal edges. In this task a baseline, two difficulty levels and one stressful condition were performed as well.

Baseline: Participants were instructed to watch the board game without touching the board itself nor the objects.Low WL: Participants were asked to remove five predefined objects (the easiest ones). They had three minutes to complete the task.High WL: Participants were asked to remove all 12 objects. They had three minutes to complete the task. When investigating stress assessment, we referred to this condition as ‘No Stress’ condition (i.e., in the comparison ‘No Stress vs. Stress’) as it differed from the Stress one only in the presence of the stressors whilst the difficulty level was the same.Stress: Participants were asked to remove all 12 objects. They had one minute to complete the task. Additionally, high intensity noise was played (85 db) and the white-coat effect was used to stress the participant [35]. Then, a four-minute relaxing music and video was played at the end of this phase. This was done to let participants recover from the stressful event before continuing with the experiment.

In all conditions, behavioral data like number of objects removed and accomplishment time were collected. The Low WL and the High WL conditions were performed randomly while the baseline and the stress conditions were performed respectively at the beginning and at the end of the experimental task. Before the baseline the participant performed a training session by extracting a couple of objects from the board.

#### 2.2.3. Webcall Task

This task consisted in an interactive Webcall to simulate a teleconference in smart-working condition. This task comprised a baseline, a positive, and a negative condition of two minutes each. The positive and negative conditions were achieved by asking the participant to respectively recall the happiest and the saddest memory of their past, while during the baseline condition the participant was asked to watch the teleconference platform interface without reacting. The positive condition was always performed at the beginning to avoid transients due to strong negative memories. One experimenter was sitting in another room interacting with the participant. The hypothesis was that asking the participant to talk about saddest/happiest memories will naturally induce these emotions and thereby enable them to feel and display the relevant expressions of emotions via multiple modalities, including physiological reactions [37,38].

### 2.3. Performance Assessment

Participants’ performance was assessed for NB and DG tasks. Webcall task did not imply a right or wrong response therefore no performance was computed. Performance in NB was assessed using the Inverse Efficiency Score (IES) [34] computed as reported in Equation (1):(1)$$\mathrm{IES}=\frac{\mathrm{RT}}{1-\mathrm{PE}}$$
where RT is the participant’s average (correct) reaction time within the condition considered, and PE is the participant’s proportion of errors in the same condition. IES can be considered as the RT corrected for the amount of errors committed [34].

For the DG task we combined the number of errors, number of extracted objects, and the time spent to complete the task, in order to have an overall value representing the performance. Since no standard Performance Index (PI) are reported in the literature, we proposed the following one:(2)$$PI=\frac{\frac{\mathrm{OBJ}}{{\mathrm{OBJ}}_{\max}}+\left(1-\frac{\mathrm{ERR}}{\mathrm{TIME}}\right)}{2}$$
where OBJ is the number of extracted objects, OBJ_max_ is the total number of objects in the condition (5 in the low WL condition and 12 in the high WL and stress ones), ERR is the maximum number of errors a participant could make in the condition (one error per second, 180 in Low WL and High WL conditions and 60 in Stress one) and TIME is the time the participant spent to complete the task in the condition.

### 2.4. Subjective Reports

After each experimental condition, including the baseline, two questionnaires were administered to the participants:NASA Task-Load Index (NASA-TLX): It consists of six sub-scales representing independent groups of variables: mental, physical and temporal demands, frustration, effort and performance [39]. The participants were initially asked to rate on a scale from “low” to “high” (from 0 to 100) each of the six dimensions during the task. Afterwards, they had to choose the most important factor along pairwise comparisons. The NASA-TLX was selected for subjectively quantify the mental demand perceived by the participants with respect to the experimental condition of DG and NB tasks.GENEVA Emotion Wheel (GEW): It is a validated instrument to measure emotional reactions to several stimuli [40]. The participants were asked to indicate the emotion he/she experienced by choosing intensities for a single emotion or a blend of several emotions out of 20 distinct emotion families. Given the nature of the task, in this analysis we decided to use only the type of emotions selected by participants, without considering their intensities.

The reason why we selected these questionnaires is because they have been adopted in several studies. In particular, the NASA-TLX has been used for WL [41,42] subjective reports and GENEVA has been used for emotion categorization [40,43]. For the stress self-report, we utilized only the temporal demand and frustration parameters because they are the main components of the stressor used in this study.

### 2.5. EOG Recording and Analysis for Mental Workload Assessment

The vertical EOG pattern was estimated by acquiring simultaneously the EEG Fpz channel of the BeMicro (EB Neuro, Florence, Italy) and the EEG TP9 channel of the Muse 2 (Interaxon Inc, Toronto, OH, USA), with a sampling frequency of 256 Hz and 64 Hz respectively. Details are summarized in Table 1. The aim of the EOG analysis was to detect the eye blinks in order to estimate the eye blink rate (EBR) and finally correlate it with the WL variations) [7,44]. The same algorithm was adopted for the analysis of both datasets. Firstly, the EOG signal was band-pass filtered using a 5th-order Butterworth filter within the frequency range of 2–10 Hz, since in this range the main frequency contribute of eye blinks is contained [45,46].

Secondly, the eyes open condition was used to identify a threshold for each participant that, when exceeded, identified a potential blink. The threshold was calculated as follows:(3)$$\mathrm{Threshold}=\mathrm{mean}\left(\mathrm{EOG}\;\mathrm{Eyes}\;\mathrm{Open}\right)\;+3*\mathrm{robustStdDev}$$
where robustStdDev is the mean absolute deviation of the corresponding EOG channel.

Finally, every time the EOG signal exceeded the computed threshold, the Pearson correlation between a common blink template (the template was built averaging the blinks estimated from five random participants during the eyes open condition) and the EOG signal was computed within each experimental condition (i.e., pattern-matching phase). If this value was higher than 0.9, a potential blink would be classified as “real blink”, similarly to what performed by the BLINKER algorithm [47].

The EBR estimated for each participant in each condition were calculated as the total number of blinks in every condition divided by the condition duration. EBR was evaluated under the different WL conditions to assess if it could differentiate user’s mental workload. Previous studies demonstrated the capability of this parameter in estimating WL demand [16,44,48].

### 2.6. EDA Recording and Analysis for Stress Assessment

The EDA was recorded by both laboratory and consumer wearable devices. The sampling frequency of the Shimmer3 GSR+ unit (Shimmer Sensing, Dublin, Ireland) laboratory device was 64 Hz while the sampling frequency of the Empatica E4 was 4 Hz. Shimmer sensors were placed on the participant’s no-dominant hand on the second and third fingers. In Empatica E4 the two electrodes are placed on the bottom part of the wrist. The EDA was firstly low-pass filtered with a cut-off frequency of 1 Hz and then processed by using the Ledalab suite [49], a specific open-source toolbox implemented within the MATLAB (MathWorks, Natik, Massachussets) environment for EDA processing (details in Table 1). The continuous decomposition analysis [50] was applied in order to estimate the tonic (SCL) and the phasic (SCR) components [51]. The SCL is the slow-changing component of the EDA signal, mostly related to the global arousal of the participant. On the contrary, the SCR is the fast-changing component of the EDA signal usually related to single stimuli reactions. The EDA components, as well as the other neurophysiological parameters, were estimated both using a 60 s time resolution and averaging within each experimental condition. Finally, only the SCL was analyzed accordingly with the objectives of the present study as demonstrated by Borghini et al. [7]. This parameter was chosen for stress estimation since previous studies demonstrated its relation with this mental state [7,52].

### 2.7. ECG Signal Recording and Analysis for Emotional State Assessment

Additonally, the HR estimation was performed using laboratory and consumer wearable technologies. ECG signal was collected using an electrode fixed on the participant’s chest (laboratory device BeMicro) and referred to the potential recorded at both the earlobes with a sampling frequency of 256 Hz. At the same time, photoplethysmographic signal (PPG) was collected by means of Empatica E4 (Empatica, Milan, Italy). First, the ECG and PPG signal were filtered using a 5th-order Butterworth band-pass filter (1–1 Hz, and 1–4 Hz, respectively) in order to reject the continuous component and the high-frequency interferences, such as that related to the mains power source (details in Table 1). Another purpose of this filtering was to emphasize the QRS process of the ECG signal [53]. The following step consisted in computing the ECG (PPG) signal to the power of 3 to emphasize the heartbeat peaks, as they generally have the highest amplitude, and at the same time reduce spurious artifact peaks. Finally, the distance between consecutive peaks (i.e., each R peak corresponds to a heartbeat) was measured to estimate the HR values every 60 s. The Pan-Tompkins algorithm [54] was used for the HR estimation. A combination of HR and SCL measurements was adopted in order to estimate EmS [55,56]. In this regard, an Emotional Index (EI) was defined as:(4)$$\mathrm{EI}=\;\left|\mathrm{SCL}\right|\;*\mathrm{HR}$$
where SCL and HR were normalized by subtracting the corresponding baseline and dividing by the corresponding standard deviation. The resulting values were then averaged within the considered experimental condition. The combination of these two parameters was adopted because the sensitivity of this emotional index was already described in previous works [56].

### 2.8. Statistical Analysis

Statistical analyses were performed after normalizing each data condition with the corresponding task Baseline. For each participant, EBR, SCL, and HR data collected during baseline were subtracted from data collected during experimental conditions. The new EBR, SCL and HR values were named respectively EBR’, SCL’ and HR’. The Shapiro–Wilk test was used to assess the normality of the distribution related to each of the considered parameters. If normality was confirmed, Student’s t-test would have been performed to pairwise compare the conditions (e.g., ‘Low WL vs. High WL’, or ‘laboratory device vs. wearable device’). In case of non-normal distribution, the Wilcoxon signed-rank test was performed. In case of comparisons between three or more distributions, the analysis of variance (ANOVA) or its non-parametric equivalent (Friedman ANOVA) was performed. For all tests, statistical significance was set at α = 0.05.

Pearson’s repeated measure correlation (rmcorr) analysis [57] was then used to assess the reliability of the parameters estimated by the wearable device with respect to the laboratory one both at single- participant level and on the entire group. The rmcorr was performed on the average values of each parameter of wearable and laboratory devices gathered during the entire experimental session.

## 3. Results

### 3.1. Performance

#### 3.1.1. N-back task

The Wilcoxon signed-rank test on the IES (Figure 3) revealed a significant difference between the low WL and high WL conditions (*p* < 0.001) and between the “no stress” (i.e., high WL) and stress conditions (*p* < 0.001). Furthermore, the three parameters involved in the IES computation (i.e., reaction times, wrong response, missed response) were analyzed to determine the one was most affecting the decreasing performance while executing the task. The Wilcoxon signed-rank test showed that both in high WL and Stress conditions (Figure 4) the number of missed responses increased significantly compared to the low WL condition (*p* < 0.001).

#### 3.1.2. Doctor Game Task

The Wilcoxon signed-rank test revealed that the performance index significantly decreased (*p* = 0.03) during the high WL condition compared to the low WL one (Figure 5). The same was observed during the Stress condition when compared with the no stress one (*p* = 0.02).

### 3.2. Subjective Reports

#### 3.2.1. N-back task

The Wilcoxon signed-rank test performed on the NASA-TLX demonstrated that participants perceived the High WL condition significantly more demanding (*p* = 0.02) than Low WL one (Figure 6). Additionally, at the end of the experiments they reported that the High WL condition resulted too difficult to be performed and that for this reason they did not or could not attend the task properly. Regarding the subjective stress evaluation, the combination of frustration and temporal demand parameters of the NASA-TLX was considered. These two parameters were selected accordingly with the relevant audio noise and the white-coat effect induced within the stress condition. The statistical analysis showed no significant difference (*p* = 0.4) in terms of perceived stress between no-stress and stress conditions.

#### 3.2.2. Doctor Game Task

Looking at NASA-TLX total score, participants did not perceive the High WL condition to be significantly harder than Low WL condition (*p* = 0.9). Additionally, in this task we considered the frustration and temporal demand parameters of the NASA-TLX to assess the perceived stress, and no significant difference was found between the no-stress and stress conditions (*p* = 0.8).

#### 3.2.3. Webcall Task

As showed in Table 2, during the positive condition participants rated mostly positive emotions than the negative ones. Instead, during negative conditions the rated emotions were mostly negative. However, some participants selected negative emotions during the positive calls while others positive emotions during the negative one.

### 3.3. Neurophysiological Results

#### 3.3.1. Methods comparisons

The statistical analysis revealed no significant difference in terms of EBR’ between the consumer wearable and laboratory equipment during both NB (*p* = 0.65) and DG (*p* = 0.69). Similarly, the Wilcoxon signed-rank tests on the SCL’ and HR’ showed no significant differences in terms of SCL’ (NB: *p* = 0.09; DG: *p* = 0.4) and HR’ (NB: *p* = 0.18; DG: *p* = 0.69) estimation. Correlation analysis between the neurophysiological parameter estimated with wearable and laboratory devices was performed. All the parameters were significantly correlated (*p* < 0.05). EBR estimated with laboratory and wearable devices resulted highly and positively correlated (R = 0.83, *p* < 10^−47^) (Figure 7). Correlation for SCL and HR resulted less strong but however significant. SCL correlation analysis (Figure 8) reported and R of 0.4 (*p* < 10^−6^). Finally, R value for HR correlation (Figure 9) was 0.51 (*p* < 10^−14^). To support correlation results, time dynamics of the investigated parameters acquired in a representative participant are depicted in Figure 10, Figure 11 and Figure 12.

#### 3.3.2. Mental workload

For both wearable and laboratory device the Wilcoxon signed-rank tests did not reveal significant differences (consumer wearable: *p* = 0.64; laboratory: *p* = 0.96) in terms of EBR’ when comparing high WL vs. low WL conditions.

#### 3.3.3. Stress

The Wilcoxon signed-rank tests on SCL’ parameter estimated by the laboratory device and the wearable one returned significant difference showing higher values during the stress condition (all *p* < 0.05) both for the NB (Figure 13) and DG (Figure 14) task.

#### 3.3.4. Emotional State

The Wilcoxon signed-rank test demonstrated no statistical differences (wearable: *p* = 0.1; laboratory: *p* = 0.4) in terms of EI between the positive and negative conditions.

## 4. Discussion

The objectives of the study consisted in assessing the reliability and capability of commercial wearable devices with respect to laboratory devices in estimating EBR, SCL and HR parameters and discriminating different levels of mental workload, stress, and emotional state.

### 4.1. Research Questions

Regarding the RQ1 (i.e., “Are the above-mentioned neurophysiological parameters (EBR, SCL and HR) gathered through consumer wearable devices comparable with those acquired with laboratory equipment?”), our results confirmed the feasibility to measure EBR, HR and SCL using consumer wearable devices. The parameters estimated with wearable and laboratory devices showed significant positive correlations as a demonstration that the two devices provided similar neurophysiological results (Figure 7, Figure 8 and Figure 9). Additionally, no statistical differences were observed in terms of EBR, HR, and SCL estimation between the two technologies considered (i.e., consumer wearable and laboratory). In fact, for each of the parameters considered the statistical analysis showed no significant difference in the averaged. These results support the adoption of consumer wearable devices and the relative collected metrics to disentangle complex mental and emotional events in real-life environments. This aspect leads to the RQ2 (i.e., “Are consumer wearable devices reliable in assessing different levels of several mental states?”). EBR was used as a neurophysiological correlate of WL, and the Muse 2 (wearable) and BeMicro (laboratory) devices were compared. No difference was found in terms of mental workload variation during the NB and DG task.

### 4.2. Workload Assessment

Regarding the DG task, the absence of WL changes was probably due to the fact that the High WL condition was not so hard as expected. Indeed, even if performance decreased in the high WL condition compared to the low WL one, participants did not perceive the high WL condition to be harder. It is arguable that adding more items resulted in a similar WL demand between low and high WL conditions with no difference when comparing EBR’ correlates.

Similarly, for the NB task, combining together performance and subjective reports, it could be argued that the absence of WL correlates was due to the difficulty of the task itself. In fact, NASA-TLX showed participants perceiving high WL condition to be harder than low WL one (Figure 6). However, at the end of the experimental session they reported that the High WL condition was too hard to be performed and for this reason they did not or could not attend the task properly. This finding is supported by performance analysis, where it was found number of missed responses significantly increased in high WL condition compared to low WL one (Figure 4). In this view, the absence of WL correlates could be a result of participants’ abandoning the task. Alternatively, the lack of EBR’ variations in both tasks could be motivated by EBR sensitivity. EBR parameter could be less sensitive to slight changes in task WL demand then other parameters (HR, HRV, PSD, ERP, etc.). This means that other parameters than EBR could have detected WL correlates in the same conditions. This points out directions for future works. The same paradigm could be tested using different neurophysiological correlates of WL to test their sensitivity and to support their adoption in different environments.

### 4.3. Stress Assessment

In terms of stress assessment, the SCL parameter was used as a neurophysiological correlate. The Empatica E4 (consumer wearable) and Shimmer (laboratory) evaluated an increased stress level during Stress condition compared to no stress one, both within NB (Figure 13) and DG (Figure 14) tasks. Even if stress correlates are accompanied with a decreased performance in both experimental tasks (Figure 3, Figure 5), participants were not able to perceive stress variations. In accordance with this, previous studies highlighted the limit in assessing perceived stress using subjective reports [34]. This study, therefore, confirmed the utility of using neurometrics to assess perceived stress [7]. It was also demonstrated that consumer wearable devices could substitute laboratory equipment to acquire such neurometrics. The possibility to detect stress in an obtrusive way is one of the most promising aspects of wearable devices.

### 4.4. Emotional State Assessment

Finally, regarding the possibility to discriminate between a positive EmS and a negative one using a combination of SCL and HR [55], both technologies were not able to differentiate these two conditions. Even if after positive condition participants selected mostly positive emotions (and negative ones after negative conditions), we found that after positive condition participants selected also some negative emotions and vice versa. It is arguable what arose from the two conditions was a blend of emotions, with no pure positive or negative connotations. Additionally, there is the possibility that two minutes interaction with a stranger in a simulated webcall was not enough to elicit a measurable neurophysiological change in the participants’ emotional states. As exposed, considering performance and subjective evaluations, the reason for the absence of WL and EmS correlates could be the experimental design itself, which did not elicit the desired mental states. This limit points out direction for next works. Future studies should design an experiment to more accurately define WL and EmS conditions.

### 4.5. Limits and Future Directions

Although both the reliability of consumer wearable devices in estimating neurophysiological signals and their capability in discriminating different levels of stress is promising, some limitations must be discussed. An experimental design and tasks capable of eliciting the desired levels of the mental states must be implemented to better investigate the usability of wearable devices. For NB and DG, an improved design should elicit the proper level of workload while for the emotional state evaluations a longer duration of the task should be considered in order to elicit a stronger and measurable emotional, and therefore autonomic, response in the participants. Consumer wearable devices are user-friendly and non-invasive technologies, allowing their usage in dynamics condition in which laboratory equipment would not be adequate. The possibility to use these devices in dynamics environments must be supported by a good quality of the gathered signals. This is a challenging aspect for consumer wearable devices and their utilization must be carefully evaluated considering the recording settings and protocol in order to acquire a valid signal. In particular, after this preliminary evaluation of wearables reliability, their capability in differentiating between different mental states should will be tested in real-working conditions with attention to the processing and analysis of the data gathered with these devices and the results will be considered for the next study. Additionally, it should be underlined that one of the considered consumer wearable devices, the Empatica E4, can be classified as a high-level wearable device. The elevated cost of high-quality wearables could represent a limit in their adoption. For this reason, the possibility to estimate the considered signals and the related mental states using commercial and low-cost wearable devices should be also explored in order to broad the mental state monitoring in the consumer world, without limiting their adoption to the scientific research.

Furthermore, future works should investigate these and other mental states in a larger group of participants and investigate the impact of participants’ movements on the quality of collected data with a particular attention to the devices/parameters affected by the movements and the intensity of the considered signals. Specifically, an important aspect that will be investigated in the next study is the comparison of the number of artifacts and the percentage of data loss found in consumer wearable devices with those of laboratory equipment. Additionally, reliability of investigated parameters in estimating mental states correlates in working-like tasks should be compared to other physiological signals (such as EEG and HRV) in order to detect the one that better fits to the recording conditions. Consequently, the adoption of other physiological signals must be accompanied by an adequate task duration to provide reliable data. Once reliability of wearable devices has been confirmed, the possibility to discriminate mental states in real-time must be investigated. Finally, consumer wearable devices are optimal candidate for health and well-being monitoring [58,59]. When appropriate algorithms are applied it is possible to monitor patients’ health by remote in real-time and prevent fatal and non-fatal occurrences. For this reason, it is important to investigate the acceptance of this wearable devices and their easiness to use [60]. This will be especially important for monitoring elderly population [61].

## 5. Conclusions

The study demonstrated that signal recorded with consumer wearable and laboratory devices showed a statistically positive correlation and no significant difference (RQ1). Additionally, it was demonstrated the capability in differentiating stress levels (RQ2). Within this experimental design it was impossible to differentiate between different levels of WL and EmS (RQ2).

The possibility to measure neurophysiological parameters at the same level laboratory devices do but with a limited invasiveness is one of the greatest points of strength of consumer wearable devices. On the other side, unobtrusiveness is achieved with reduced size which comports a limited duration of the battery, limiting these devices to short periods of testing. Furthermore, it is reported that the contact between wearable devices and the body id not always optimal, leading to missing or altered data [25]. This limits the use of consumer wearables to those case in which movement is compatible with data collecting.

Taken together, these findings support the adoption of low-cost wearable device to monitor operators’ mental states in laboratory and real-life environments. The possibility to unobtrusively assess mental states has broad applications. It could be possible to monitor air-traffic controllers, medical operators, surgeons, while working without interfering with the performance. Hopefully, the ability to better differentiate between mental states will reduce the effect of tragic occurrences.

## Figures and Tables

**Figure 1 sensors-21-02332-f001:**
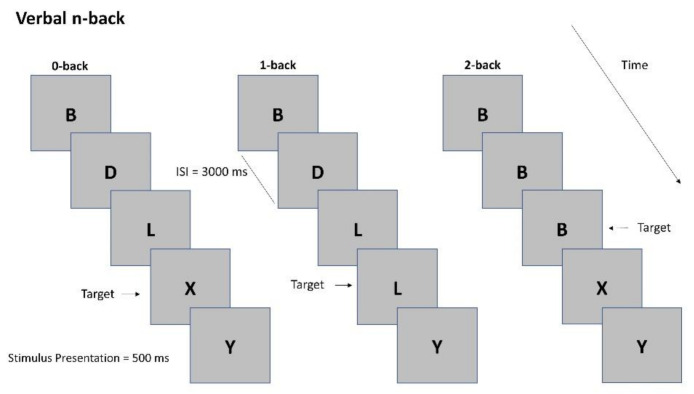
Example of N-back task under the 0-back, 1-back, and 2-back conditions.

**Figure 2 sensors-21-02332-f002:**
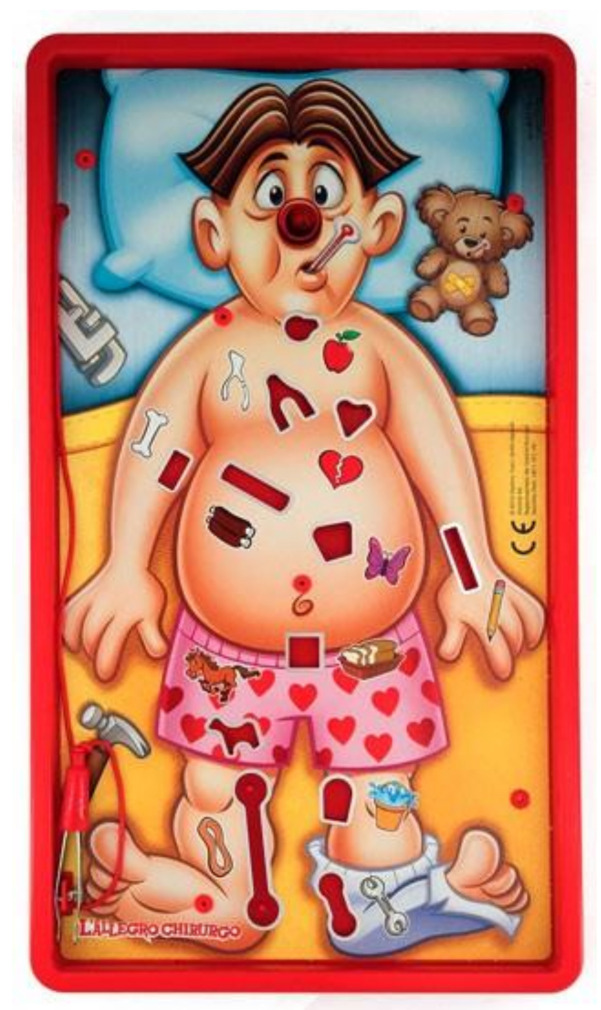
The Doctor Game task consisted in extracting as many objects as possible from the “patient” without touching the metal border. If an error occurred, the nose will emit a red light and the board will vibrate.

**Figure 3 sensors-21-02332-f003:**
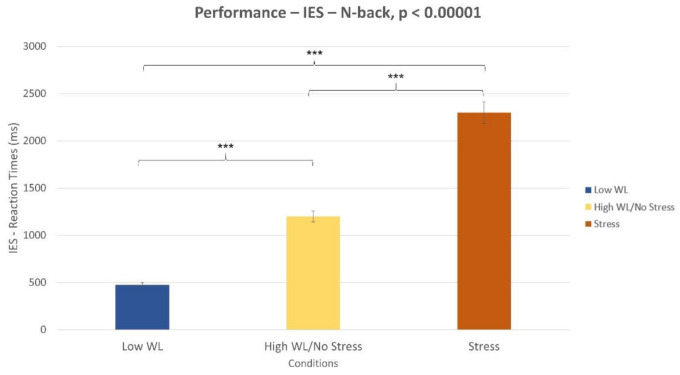
Difference in subjective performance during N-back task. Low vs. high Workload (WL) conditions (*p* < 0.001). No stress vs. stress conditions (*p* < 0.001).

**Figure 4 sensors-21-02332-f004:**
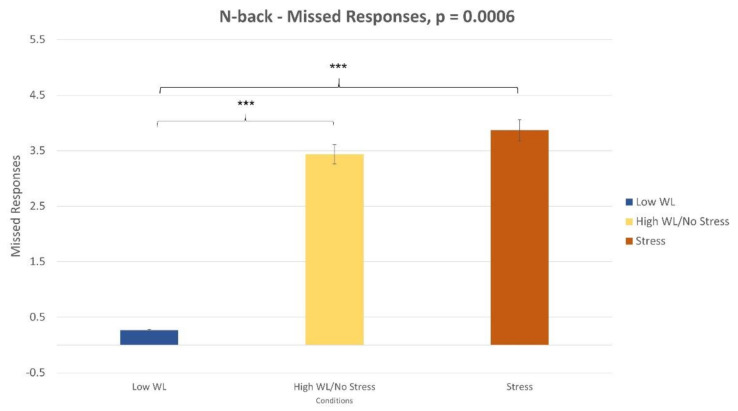
The number of missed responses was higher in high WL and stress conditions compared to the low WL condition (*p* < 0.001).

**Figure 5 sensors-21-02332-f005:**
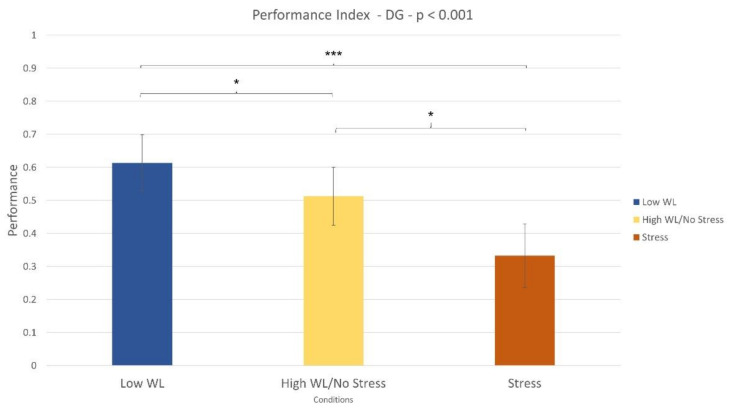
The performance index significantly decreased during the high WL condition compared to Low WL condition (*p* = 0.03). The same result was found in the stress vs. no stress comparison (*p* = 0.001).

**Figure 6 sensors-21-02332-f006:**
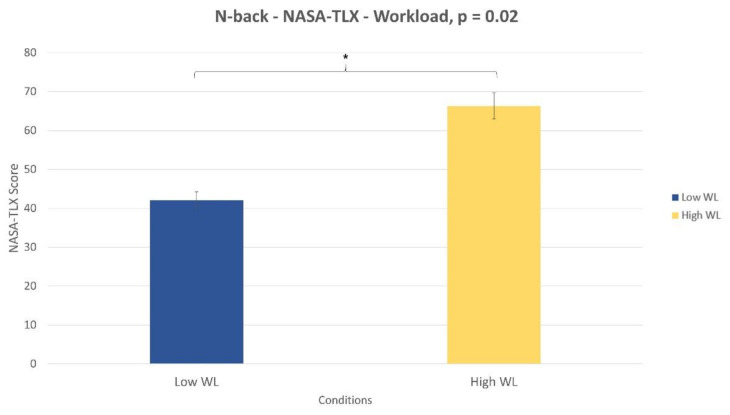
NASA-TLX total score during the low WL and high WL conditions (*p* = 0.02).

**Figure 7 sensors-21-02332-f007:**
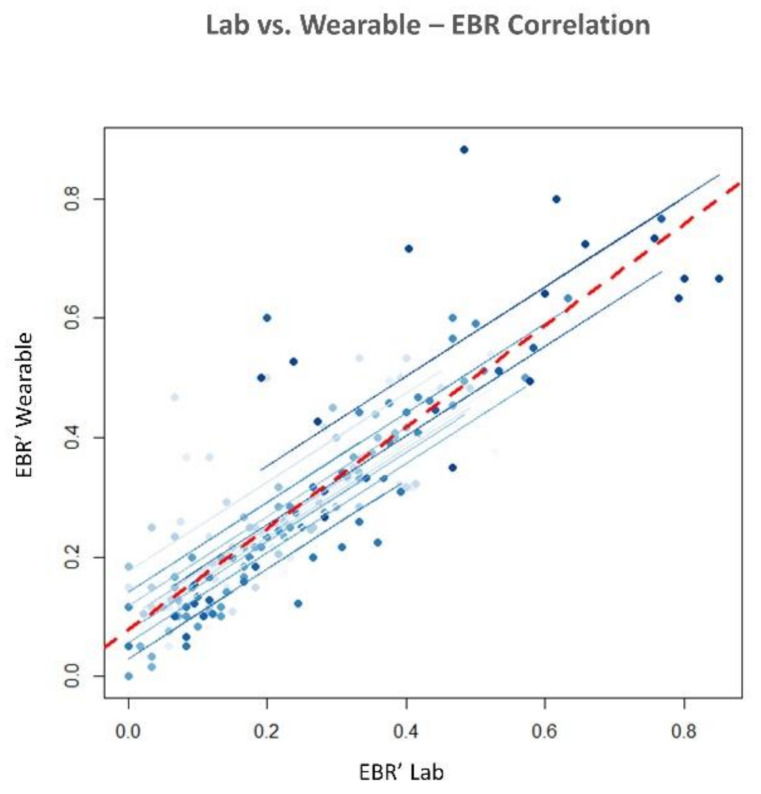
Pearson’s repeated measure correlation for the Eyeblink Rate (EBR) estimated with laboratory and wearable devices. R = 0.83, *p* < 10^−47^.

**Figure 8 sensors-21-02332-f008:**
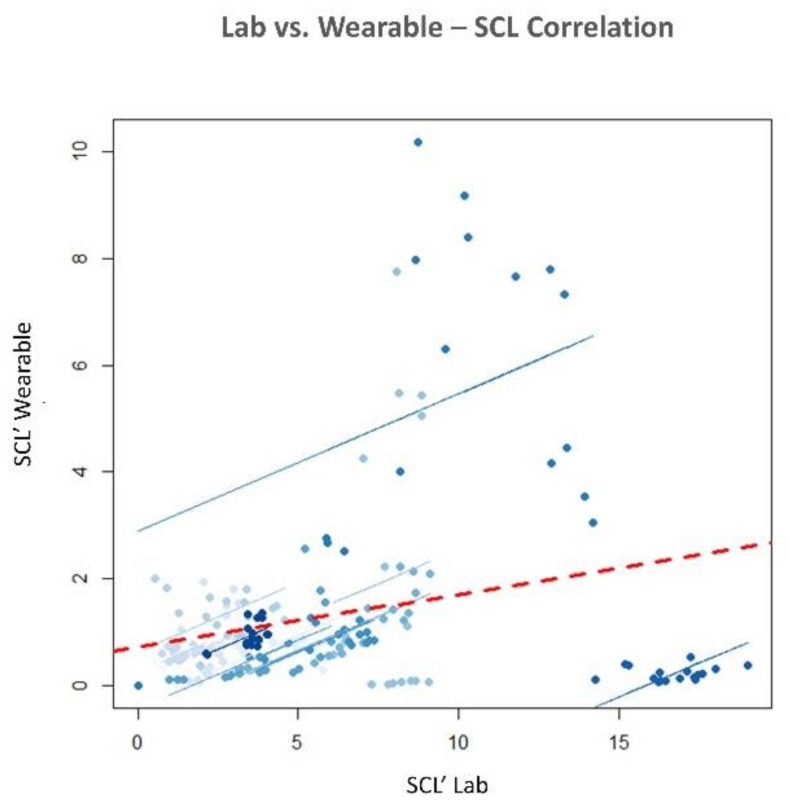
Pearson’s repeated measure correlation for the Skin Conductance Level (SCL) estimated with laboratory and wearable devices. R = 0.4, *p* < 10^−6^.

**Figure 9 sensors-21-02332-f009:**
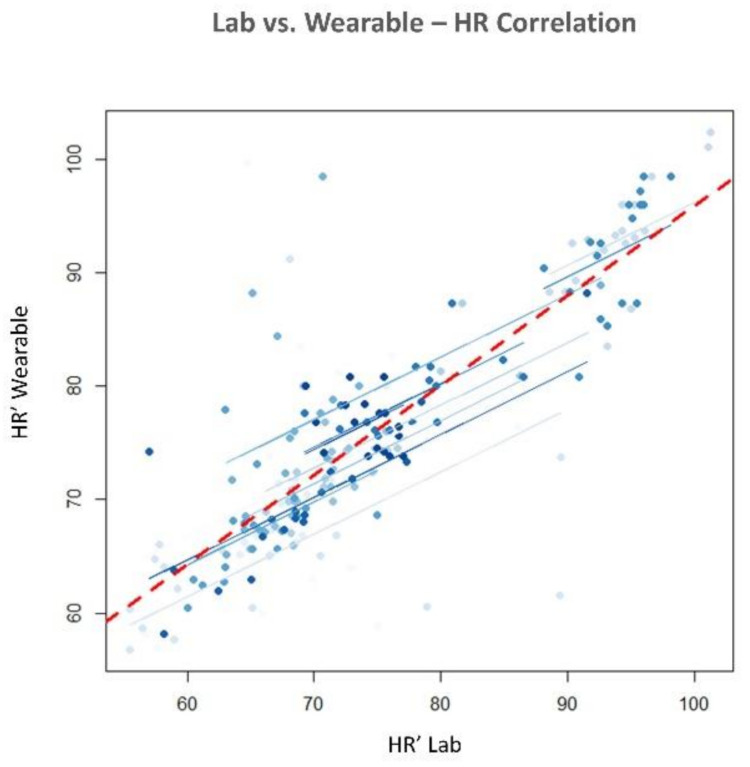
Pearson’s repeated measure correlation for the Heart Rate (HR) estimated with laboratory and wearable devices. R = 0.51, *p* <10^−14^.

**Figure 10 sensors-21-02332-f010:**
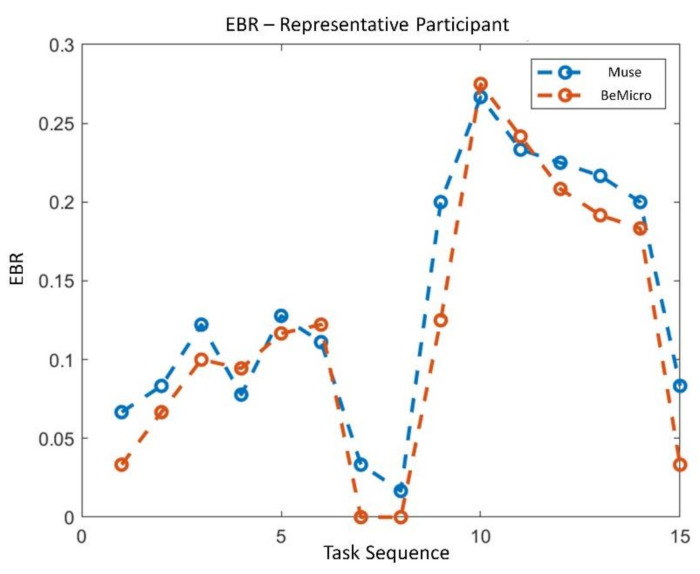
Time dynamics of EBR across all experimental task and conditions for both consumer wearable (blue) and laboratory device (red).

**Figure 11 sensors-21-02332-f011:**
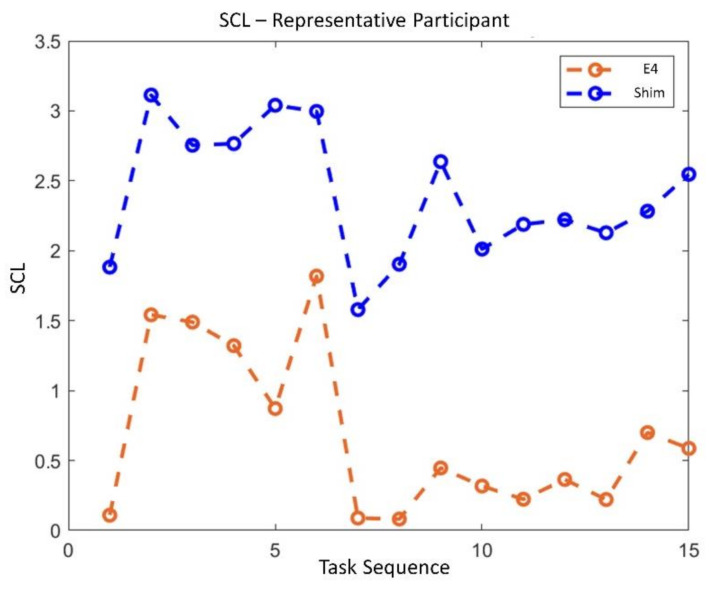
Time dynamics of SCL across all experimental task and conditions for both consumer wearable (red) and laboratory device (blue).

**Figure 12 sensors-21-02332-f012:**
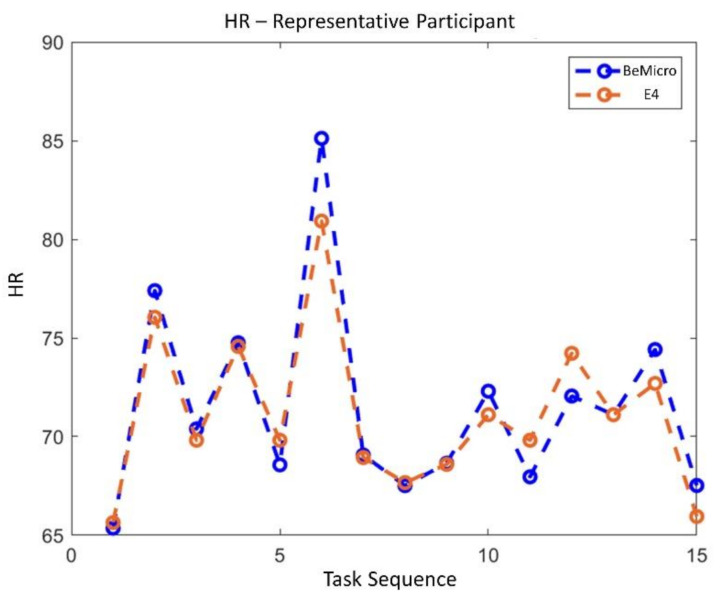
Time dynamics of EBR across all experimental task and conditions for both consumer wearable (red) and laboratory device (blue).

**Figure 13 sensors-21-02332-f013:**
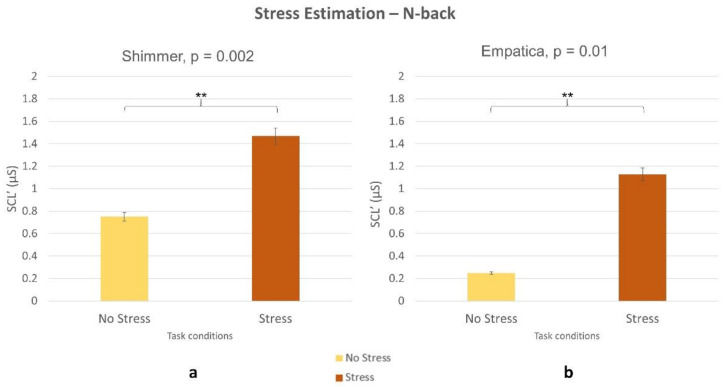
Increased SCL’ in stress vs. no stress condition during NB task. Statistical analysis revealed significant difference between the conditions for both (**a**) laboratory equipment (*p* = 0.002) and (**b**) wearable device (*p* = 0.1).

**Figure 14 sensors-21-02332-f014:**
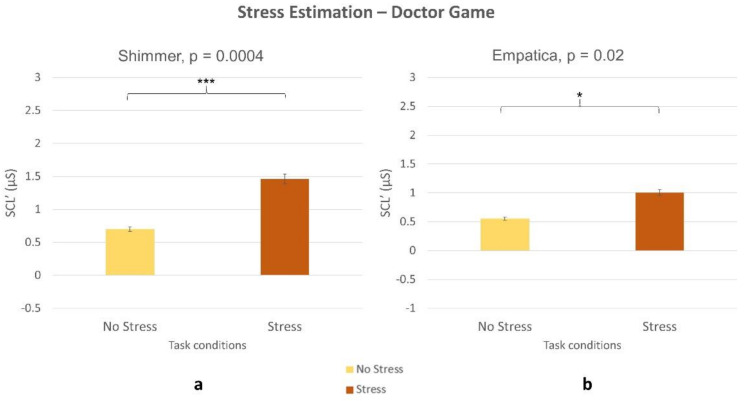
Increased SCL’ in stress vs. no stress condition during DG task. Statistical analysis revealed significant difference between the conditions for both (**a**) laboratory equipment (*p* = 0.0004) and (**b**) wearable device (*p* = 0.02).

**Table 1 sensors-21-02332-t001:** A summary of the devices and signals used in the presented work.

Signal	Laboratory Device	Consumer Wearable Device	Extracted Feature	Filter Frequency Range	Time Window
EOG	BeMicro	Muse 2	EBR	2–10 Hz	-
EDA	Shimmer	Empatica 4	SCL	1 Hz	60 s
PPG	-	Empatica 4	HR	1–4 Hz	60 s
ECG	BeMicro	-	HR	1–15 Hz	60 s

**Table 2 sensors-21-02332-t002:** Frequency of the emotions selected after positive and negative conditions of the Webcall.

Emotions (Geneva Emotion Wheel)	Positive Webcall	Negative Webcall
Admiration	1	
Contentment	1	1
Joy	12	
Love	3	2
Pleasure	6	
Pride	3	1
Relief		1
Interest	6	2
Embarrassment	1	
Compassion		1
Anger	1	2
Disappointment		4
Disgust		1
Fear		3
Guilt		3
Regret	1	1
Sadness	2	11
Shame	1	3

## Data Availability

The aggregated data presented in this study might be available on request from the corresponding author. The data are not publicly available because they were collected within the EU Project “WORKINGAGE: Smart Working environments for all Ages” (GA n.826232) and they are property of the Consortium.

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
