# Peer review of "Wearable Technologies for Mental Workload, Stress, and Emotional State Assessment during Working-Like Tasks: A Comparison with Laboratory Technologies"

_sensors, 2021, doi:10.3390/s21072332_

Round 1

Reviewer 1 Report

The authors presented, in a very well organized manuscript, the results  of a study  where they tried to answer in two research questions, named RQ1 and RQ2.

I think that  the results are sufficient to support the hypotheses of the work, but still have  some questions to them which need their answers.

  1. Wishing to study real life problems they are using wearable systems to evaluate WL, Stress and ES, but the tests are simple laboratory ones, with no connection with real working conditions. Why and how they will close the gap between lab tests  and real life  working environments?
  2. The authors claim, correctly, that mental and cognitive states play  an important role in the avoidance of fatal errors in work and that monitoring of vital signals and the biomarkers extracted from them can be used to alert the crossing of the dangerous thresholds.   In my opinion the results provide  strong statistical support in a static manner.  I did not  see any reference to the  time variation of the biomarkers which then can be used for real time monitoring and consequently  for  real time alert signals.
  3. I would like to see comments from the authors why they excluded  from the study the analysis of raw EEG signals  (not only ERPs) where they are in the recent literature plethora of works where EEG is used for mental state and cognitive level monitoring.

Minor comments

  1. The Performance Index (PI) in Eq2 is a prototype equation or has been defined previously? Please provide details.
  2. Usually the value of a threshold is defined as Mean Value + n* Standard Deviation with n=1,2,3…  with corresponding confidence e.g. 90%, 95% , 99% etc. Is Equation 3 arbitrarily defined or not? Please provide a complete answer.
  3. In the  equation ?? = |???| ∗ ?? the multiplication of the two parameters increases the sensitivity of the Emotional Index (EI) but also the noise level in the signals? Have you studied the role of the  two parameters for an Emotion Estimator  as independent variables and found that the  Eq (4) is better?
  4. It is known that the emotional mechanisms on EDA and HR are not identical. Please explain why is better to use them in one formula (eq 4).
  5. Please provide quantitative information in a  tabulated form the engineering terms of signals used in the study, e.g. spectrum, frequency range (if are more than one) and time windows  of observations etc.

Reviewer 2 Report

Dear Editor,

Thank you very much for the opportunity to review sensors-1141415:  Wearable Technologies for Mental Workload, Stress, and Emotional State Assessment during Working-like Tasks.

Overall the present study uses an interesting and useful study design. More studies like this are yet available, though remain relevant to validate wearable technologies on their technical merit. Nevertheless, some considerations with regards to the present study do hamper my initial enthusiasm. Primary they are the following:

There is a considerable percentage of (yet unexplained) missing data, making the rather small sample size even smaller. Having said that, from experience, the (remaining) sample size is typically yet sufficient to answer the proposed research questions under the present conditions. However, it might present some restrictions on the interpretation of at least some of the findings (e.g. on neurophysiological outcomes).

Wearables of course have one fundamental challenge on and above any high quality stationary (typical laboratory) equipment, being the quality and reliability of the collected data in ‘ecological valid’ environments. Meaning under movement, wide (and rather variable) temperature ranges, usage and misusage, rapid changes in values (or even ranges!) of the relevant parameters (e.g. due to aforementioned reasons), suddenly interrupted data series, battery charging, missing data, uncontrollable individual factors (skin features, naturel arrythmias, vascular and/or metabolic factors), consumer / client (non)compliance, …  these have clearly not been addressed in the present study. The considerable percentage of missing data in the present (ecological reasonable controlled) study might very well be a relevant indication from that perspective however (one can presume)? It might be a relevant consideration to reflect on (e.g. in the introduction and/or discussion).   

Minor comments:  

  • This type of technology might not only be relevant for “monitoring user’s performance”, though might very well be used (when valid algorithms are applied) for health & well-being?

  • Line 69 – 70: This generally works well within relatively low HR ranges, however above 120 bpm E4 seems to be less reliable (for example implying that if such tasks are conducted while moving, e.g. running around an office, the Empathica might run into some technical caveats).

  • Line 73 – 75 Nevertheless the Empathica E4 is rather expensive as compared to other wearable devices (fit-bit, Philips, others) and seems to offer a rather arbitrary collection of sensors / parameters (which do individually make sense, though regularly lack overall neuro-physiological coherence).

  • Out of 17 recruited participants only data of 12, that’s quite a significant loss of data, how come, what was generally the reason for missing data?

  • L185-186 “a teleconference in a Positive, and a Negative condition of two minutes each.” Social interaction tasks like this seem to induce only a marginal autonomic response (i.e. skin conductance, heart rate and its derivatives, like heart rate variability) at best. Whilst the relatively short periods are considered not extensive enough (and interviews like this, even when conducted ‘live’, not intrusive enough) to ignite the type of psychoneuroedocrinological responses (e.g. cortisol, epinephrine) typically expected from social interaction tasks (like for example the Trier Social Stress Task, TSST). More specifically, would the expected autonomic responses be sufficient enough (e.g. related to the sample size) to indeed capture (potential) significant (differences in) responses?    

  • With the additional consideration that for more sophisticated HRV analytics (e.g. FFT based) the 2 minute period is generally regarded too short to provide reliable data. A caveat that is typically bypassed in wearable technologies by using time domain parameters (like Root Mean Sqaured Successive Difference (RMSSD) of sequential inter-heart beat interval) (for wearables this should consequently not necessarily be a show stopper).

  • Line 211-228 Subjective reports: The NASA Task-Load Index (NASA-TLX) and GENEVA Emotion Wheel (GEW) are indeed interesting and promising methods, though not yet that well validated or generally adopted. We will follow there usage and development with interest, however, why did the authors currently opt for these and not alternatives?: what was their primary advantage on and above other potential self-reports on mood and performance?

  • Line 356 – 365 Webcall: which of these findings were actually significant (in a statistical sense)? If none, or not tested (including correction for multiple comparisons), may be more insightful to present in a frequency table?

  • Line 366 – 395 Neurophysiological results: here the yet to some extent ‘obscure’ results might indeed be related to the eventually relatively small sample size? What would the authors expect if more data could have been collected?, and secondly, how do the authors interpret the noted (rather contrary) deviances in the response of the Empatica E4 (e.g. figure 11 b)?

Reviewer 3 Report

  The paper presents a comparison between modern and more portable wearable devices that can be used at home and traditional lab use devices with respect to their ability to monitor and extract stress and mental workload during work-like tasks. The authors have created a protocol to simulate three different working tasks, each with five levels of stress/load. The 17 (12 valid) participants wore sets of lab/traditional and home/wearable devices extracting three different metrics to compare. They also filled in questionnaires for subjective workload and emotional states. The outcomes show that  1) both sets of devices showed significant positive correlations and no statistically significant difference 2) both sets of devices can discriminate stress levels but not work load and emotional state (as oftentimes even the participants themselves could not). The work is extremely methodic and thorough in its organization, presentation and discussion.  

The most general comment would be for the general "semantics" of wearables vs. traditional/lab devices that the authors debate. Looking especially into which devices represent each category, - "wearables" are Empatica E4 and Muse 2 - "traditional/lab" are Shimmer3 GSR+ and BeMicro Shimmer GSR and BeMicro are indeed intended for lab use, yet they are also quite modern and portable. They are also in fact "wearable" in the general sense. I suppose the actual difference here with the "lab" devices is that they can be used at "home" instead of the "lab" and that they are more "portable" and sustainable by the end-users themselves. Muse 2 in fact is a "consumer" device even. Or perhaps "wearable" could mean that it is 24/7 "wearable" which should be defined. I would suggest:

  • Instead of the authors using the terms "laboratory" and "traditional" device interchangeably, to use one term of even "lab/traditional". I am less in favor of traditional since the devices are quite modern.
  • Instead of just "wearables" for the modern portable devices that can be self-sustained to use "home/wearable".
  • To first define strictly and early on which two terms you will be using and what they mean for the context of this paper.  

After that, I would suggest critical changes to the title, abstract (and perhaps throughout the paper) to reflect the core comparison that this work performs.  

For such a considerable amount of complex and thorough work, the authors could try to help the readers by simplifying some sections: -

  • Dividing large sections into more paragraphs with cohesive meaning especially in discussion and introductions
  • Summarizing intentions and outcomes before diving into details in each paragraph, e.g., RQ2 is discussed quite a lot but yet I am not sure whether all cases are covered (all tasks and work loads) and where they are attributed to (the protocol?)  

Readers have to invest a lot to grasp which devices are compared to which, which signals are extracted and compared etc. This could be solved by:

  • adding a table with the raw signals, traditional/lab device and home/wearable device, processing and extracted features signals.

Rough example:

  • EOG: BeMicro vs Muse 2 -> EBR
  • EDA: Shimmer 3 GSR+ vs Empatica E4 -> SCL
  • ECG/PPG: BeMicro vs Empatica E4 -> HR
  • Questionnaires ... -> metrics etc.

(Please forgive any misunderstandings here. This is my rough understanding from the text, with which the table will help anyway)  

Some points for discussion:  

  • the sample although small, provides significant results. Since this is about mental workload simulating working life did their background and current occupation play a role in the tasks? e.g. an actual surgeon could be less "loaded" in the Doctor game, various professions could excel in the first game and people that are trained to handle stress better e.g., law enforcement could be less "loaded" in the webcall.  
  • The overall theme is using 24/7 wearables at home instead of lab equipment. Supposedly since this could be better accepted and sustained. Was this the motivation behind the work and how did the users respond to accepting either or both of device sets? Could they sustain the "wearable" devices themselves at home? A list of human factors for acceptance can be found here, especially for the elderly in the WorkingAge project * Stavropoulos, T. G., Lazarou, I., Strantsalis, D., Nikolopoulos, S., Kompatsiaris, I., Koumanakos, G., ... & Tsolaki, M. (2020, September). Human Factors and Requirements of People with Mild Cognitive Impairment, their Caregivers and Healthcare Professionals for eHealth Systems with Wearable Trackers. In 2020 IEEE International Conference on Human-Machine Systems (ICHMS) (pp. 1-6). IEEE.  

The paper is extremely well written with the only typo perhaps being: -EBR was used as neurophysiological -> ... a neurophysiological

Round 2

Reviewer 1 Report

I am  very pleased from your  answers. 

Author Response

Dear Reviewer 2,

thank you for your valuable support in the revision process.
Bests

Reviewer 2 Report

Dear Editor and authors,

Thank you very much for the clearly diligent work being done on revising  sensors-1141415:  Wearable Technologies for Mental Workload, Stress, and Emotional State Assessment during Working-like Tasks.

I believe the work has been significantly improved, and am happy to have been instrumental in repairing some technical/data-processing inaccuracies. It makes the study much more coherent and consequently relatively comfortable to read and understand.

Some minor concern now remain:

  • Indeed it cannot be stressed enough that studies like the present are relevant and useful, though much more studies remain necessary to particularly assess and improve the usability of technologies like the present ones (either as sensor suit and/or the accompanying data processing & analytical capabilities) in their proposed ecological environments. The present results underline the reliability and validity of the data collected (like comparable studies before), though not necessarily address such ever more relevant challenges now the ambition to obtain just as good and interpretable data out of laboratory settings rapidly matures.
  • Minor remark: for precisely the reasons mentioned in the manuscript and the responses of the authors to my initial feedback, I am not sure whether or not I would consider the Empathica E4 a ‘consumer wearable’? Following the challenges formulated in the present study, it might be considered a (very) relevant ‘intermediate technology’, a first step / effort in bringing the assessment excellence of laboratory environments in to the real, consumer, world? For this reason it seems to be primarily used by (scientific) teams doing research in more ecological valid conditions. As with all its caveats (technological, consumer behaviors, other) it might yet be significantly restricted in its ‘consumer’ potency?   

May be the authors want to consider to (shortly) address these in the Discussion.

  • Finally: On some points in the manuscript, the formatting seems to be rather ‘messy’. Though I trust the editors will address these. E.g. lines 262-264 and 291 – 297 (on page 7 of 23); and lines 308 – 315 (page 8 of 23).  

Author Response

Response to Reviewer 2 Comments

Point 1: Indeed it cannot be stressed enough that studies like the present are relevant and useful, though much more studies remain necessary to particularly assess and improve the usability of technologies like the present ones (either as sensor suit and/or the accompanying data processing & analytical capabilities) in their proposed ecological environments. The present results underline the reliability and validity of the data collected (like comparable studies before), though not necessarily address such ever more relevant challenges now the ambition to obtain just as good and interpretable data out of laboratory settings rapidly matures.

Response 1: Thank you for your observation. We totally agree with you. This study was conducted in a controlled environment and it was propaedeutic to the final phase of the WorkingAge Project, during which these devices will be tested in three real-working environments similar to those simulated in this study. We expect to test these devices in this phase and to publish the related results in a next paper. We updated the Discussion section as follow:

“In particular, after this preliminary evaluation of wearables reliability, their capability in differentiating between different mental states will be tested in real-working conditions with attention to the processing and analysis of the data gathered with these devices and the results will be considered for the next study.”

Point 2: Minor remark: for precisely the reasons mentioned in the manuscript and the responses of the authors to my initial feedback, I am not sure whether or not I would consider the Empathica E4 a ‘consumer wearable’? Following the challenges formulated in the present study, it might be considered a (very) relevant ‘intermediate technology’, a first step / effort in bringing the assessment excellence of laboratory environments in to the real, consumer, world? For this reason it seems to be primarily used by (scientific) teams doing research in more ecological valid conditions. As with all its caveats (technological, consumer behaviors, other) it might yet be significantly restricted in its ‘consumer’ potency? 

Response 2: Thank you for your comment. We highlighted the fact that Empatica E4 could be classified as a high-level wearable, with respect to the Muse 2, and that this could represents a limit in its adoption out of scientific field. We implemented the Discussion section as follow:

“Also, it should be underlined that one of the considered consumer wearable devices, the Empatica E4, can be classified as a high-level wearable device. The elevated cost of high-quality wearables could represent a limit in their adoption. For this reason, the possibility to estimate the considered signals and the related mental states using commercial and low-cost wearable devices should be also explored in order to broad the mental state monitoring in the consumer world, without limiting their adoption to the scientific research.”
